# Evaluating Ecological Impact and Sustainability in the Manufacturing of Advanced Therapies: Comparative Analysis of Greenhouse Gas Emissions in the Production of ATMPs in Open and Closed Systems

**DOI:** 10.3390/bioengineering10091047

**Published:** 2023-09-06

**Authors:** Giuseppe Pinnetta, Aloe Adamini, Franco Severina, Franca Fagioli, Cristina Zanini, Ivana Ferrero

**Affiliations:** 1Stem Cell Transplantation and Cellular Therapy Laboratory, City of Health and Science of Turin, Regina Margherita Children’s Hospital, 10126 Turin, Italy; gpinnetta@cittadellasalute.to.it (G.P.); aadamini@cittadellasalute.to.it (A.A.); franca.fagioli@unito.it (F.F.); 2BIOAIR S.p.A. Scientific Department & Training Centre, Molecular Biotechnology Centre, University of Turin, 10126 Turin, Italyc.zanini@bioair.it (C.Z.); 3Department of Public Health and Pediatrics, University of Turin, 10126 Turin, Italy

**Keywords:** ATMP, sustainability, open system, closed system, greenhouse gases

## Abstract

The primary aim of this systematic analysis is to highlight opportunities to improve the environmental impact of advanced therapy medicinal products (ATMP) manufacturing. We have compared the Greenhouse Gas (GHG) emissions expressed in CO2eq of a classic clean room open system (AinB) Cell Factory versus a comparable closed system equipped with isolators (AinD). We have therefore outlined a theoretical situation to simulate the use of a closed system with an equivalent production output to that obtained in the Cell Factory (CF) of the Regina Margherita Children’s Hospital. Open and closed systems for ATMPs have been compared as regards energy requirements, ecological footprints, and costs by analyzing a hypothetic cell production cycle of 21 days. The results demonstrate energy saving and a reduction of 52% in GHG emissions using closed systems per process cycle. Moreover, a reduction in production costs in an isolator setting is also evident. This study shows that the closed system solution has evident advantages compared with the open one.

## 1. Introduction

The European Regulation 1394/2007 defines advanced therapy medicinal products (ATMPs) as products utilized to restore, correct, or modify physiological functions primarily through a pharmacological action [1]. ATMPs hold significant potential for a wide range of medical conditions and have the capacity to benefit millions of patients. These products are categorized into four primary groups: -Gene therapy medicinal products: they consist of a recombinant nucleic acid able to induce a therapeutic, prophylactic, or diagnostic effect designed to modify, control, inhibit, or overexpress a specific target gene;-Somatic cell therapy medicinal products: they include cells or tissues manipulated to alter biological characteristics, physiological functions, or structural properties with the aim of treating, preventing, or diagnosing diseases. They may be autologous, allogeneic, or xenogeneic;-Tissue-engineered medicinal products: they are composed of cells or tissues that have been modified with a combination of different techniques of cellular and molecular biology, biomaterials, and engineering to be used to repair, regenerate, or replace human tissues;-Combined advanced therapy medicinal products: they contain one or more medical devices as an integral part of the cell- or tissue-based medicinal product [2,3].

The ATMP manufacturing process for clinical use should comply with the principles of Good Manufacturing Practices (GMP) to determine how cell preparations are produced and controlled, from the collection and manipulation of raw materials to the processing of intermediate products and quality controls, storage, labelling, packaging, and release [4].

Annex 1 of the GMP [5] describes the appropriate classification required and assigned to the pharmaceutical production rooms, depending on the type of operations taking place in the facility:-Grade D: represents a clean area where the less critical stages of sterile production take place. This zone requires a specific air and surface quality, in terms of particulate and microbiological contamination, obtained by an appropriate air filtration system. Temperature and relative humidity control is also required.-Grade C: this zone is accessed from Grade D and is associated with a clean area meant for less critical steps in the manufacturing of sterile products. Grade C needs greater particle and microbiological air and surface quality compared to Grade D, while it shares similar temperature and relative humidity control.-Grade B: for aseptic preparation and filling, this is the background environment for Grade A. Air pressure differences should be continuously monitored. It requires higher particle and microbiological air and surface quality compared to Grade C, sharing a similar control of humidity and temperature. The conditions of environmental contamination must be continuously monitored.-Grade A: an area of high-risk operations that require asepsis (i.e., filling, opening, and closing of containers, such as vials or test tubes). These conditions are usually guaranteed by a homogeneous laminar air flow in the work environment, and they are monitored for the entire duration of critical processes [3].

A typical Grade B area is the so-called “clean room”, where, thanks to high-efficiency particulate air filters (HEPA) and a controlled Heating, Ventilation, and Air Conditioning (HVAC) system, the number of airborne particles is controlled and classified.

ATMP processing is an important technology, with strong companies focusing significant efforts on developing new approaches to reducing costs and improving patient safety. However, only a few approved biological therapies are available, due to economic and organizational aspects [6]. Moreover, ATMP manufacturing includes quality controls and validation processes that could be challenging when applied to personalized ATMPs of small volumes and short production times [7].

Due to its enormous complexity, ATMP production presents quite high costs, which mainly consist of direct costs, such as reagents, medical devices, materials, and products, clothing, personnel, and costs related to quality controls. However, this production has a significant impact on indirect costs as well, which are related to the qualifications and validations (i.e., media fill, cleaning, and environmental controls) and to the entire structure, including energy costs, heating costs, HVAC plants, and environmental validation. Unfortunately, their diffusion is limited due to the high production costs and complex processes in sterile environments. “Open system” Cell Factories, manufacturing ATMPs in Grade B background environments with Grade A Biological Safety Cabinets, are typically the most resource-intensive areas of a hospital, as they are five to eight times more energy-intensive than the rest of the hospital and a major contributor to waste (Figure 1A).

A typical Grade B clean room for clinical application requires high running and maintenance costs, trained operators, and well-defined procedures to prepare the rooms and the people involved in the processes. While today, production mainly occurs in open systems, there is evidence of processes in closed systems, such as isolators [6].

This is one of the reasons why, most recently, in 2017, a specific Part of EudraLex Vol. 4 was published where the production of ATMPs in “closed systems” was outlined. Closed systems, such as, for example, isolators, bioreactors, and other devices classifiable as “closed systems” are “physical barriers” which, as indicated in the EudraLex Vol. 4 Part IV guidelines, can be used to produce ATMPs (Figure 1B). Thanks to this strict physical separation between operators and the product, the use of closed systems can be performed within a Grade D background area, as stated in Part IV (§9.5.1): “[ATMPs] Manufacturing should take place in clean areas of appropriate environmental cleanliness level. More specifically, as regards production in a closed system, in an isolator, or positive pressure isolators: a background clean area of Grade D is acceptable” [8].

An isolator is an isolated self-standing environment where gloves are affixed to the enclosure surrounding the critical zone. Air within the isolator is high-efficiency particulate air (HEPA) filtered using air drawn from the room. The isolator maintains a positive pressure at all times and is never opened during use. The isolator, therefore, is a Grade A, aseptic closed system, and the characteristics of the room in which to install a production isolator (Grade A) are those indicated in Annex 1, which requires a controlled environment and at least a Grade D environment in the case of sterile production [6].

It is quite intuitive that the use of environments classified as Grade D, allowed by the GMP Guidelines when Grade A isolators are utilized for ATMPs manufacturing, with a strict and physical separation between the product and operators, substantially reduces energy costs and the environmental impact of the Cell Factory, and therefore reduces its ecological footprint. 

Moreover, other important advantages obtained when using closed system solutions, compared to open systems, can be summarized as follows:-Better protection of the product against what constitutes the greatest risk of contamination; that is to say, the operators.-Better protection of operators against potentially dangerous compounds, such as viral vectors in the case of gene therapies.-Higher level of guarantee of sterility than that obtained in open systems, thanks to the small volumes of the isolators compared to the sizes of Grade B clean rooms.-Lower risk and easier risk assessments with isolators than with open system clean rooms.-Cost reductions in infrastructure manufacturing, staff, gowning, validation, decontamination, and other costs associated with open systems, resulting in a lower Total Cost of Ownership (TCO) overall.-Higher sustainability and lower environmental impact.

In this study, we have analyzed the sustainability of a classic clean room open system (AinB) Cell Factory vs. a comparable closed system equipped with isolators (AinD) in terms of Greenhouse Gas (GHG) emissions, expressed in CO2eq, and costs, to highlight opportunities for improving the environmental impact of ATMPs manufacturing. We have therefore outlined a theoretical situation to simulate the use of a closed system with an equivalent production output to that obtained in the CF of the Regina Margherita Children’s Hospital. 

Based on these layouts, we have analytically calculated the energy savings and, therefore, the environmental impact expressed in CO2eq for GHG emissions, thus determining the total carbon footprint for this configuration.

## 2. Materials and Methods

The analysis of the difference between open and closed systems was performed by comparing the Regina Margherita CF and the Centre de Production Cellulaire (CPC) based in Epalinges and belonging to the Lausanne University Hospital (CHUV).

The Cell Factory (CF) of the Regina Margherita Children’s Hospital is part of the City of Health and Science of Turin, and it forms part of the Paediatric Onco-Hematology Division, the scientific center of excellence for the treatment of tumors from childhood to adolescence. The hospital also hosts the university, providing highly specialized services in the field of diagnosis and treatment. The CF includes a laboratory for production, involved in the extensive manipulation of ATMPs, and a Quality Control (QC) laboratory, which deals with the execution of control tests in process and batch release and environmental microbiological checks. Furthermore, a warehouse section, which depends on QC, is transversal to all the activities and manages inventory, the loading and unloading of materials, and administrative procedures for supplies.

The production area of the CF has been designed and qualified according to GMP standards [4]. It includes three Grade B laboratories (L), named L2-L3-L4 (Figure 2), each equipped with a Grade A Biological Safety Cabinet, one or two incubators for cell culture maintenance, a centrifuge, and a microscope. The Grade D room (L1) is dedicated to the sorting of samples and the storage of reagents and materials. There are two gowning corridors (entrance and exit) and one Grade C corridor. Interlocking doors regulate the clean/dirty flows of personnel and materials. Each room is equipped with a pass-box for the passage of raw materials, reagents, and finished batches.

The QC laboratory has two rooms, with not classified environments, independent staff, and material flows: the laboratory for flow cytometry activities, LAL tests, and refrigerated materials storage (L6), and the laboratory for microbiology and cell culture activities (L7).

The data relevant to the Regina Margherita open system CF (AinB) are real and measured data; all the necessary information concerning the installed power of the laboratory equipment, the volume and air changes/hour, the daily energy consumption, etc., were obtained with the support of the Hospital Technical Department. The cost data were collected from Biomanagement (SOL S.p.a., Monza, Italy), a validated software in compliance with GAMP 5, which allows the tracing and the management of all the materials and products (drugs, reagents, and medical devices) through the production process.

For the closed system equivalent layout (AinD), the data have been collected from those relevant to the CPC (Centre de Production Cellulaire) based in Epalinges and belonging to the Lausanne University Hospital (CHUV), as described by Chemili and collaborators [9].

As seen above, the CF is equipped with three Grade B clean rooms with a 120 cm Biological Safety Cabinet (BSC) in two of them (L2-L3) and a 180 cm BSC in the third (L4), plus other equipment needed for the production cycle. 

Consequently, we held that the equivalent and comparable closed system should include three Grade D clean rooms, two of them equipped with an isolator with a two-glove working area and one with a four-glove working area (Figure 3). Identical types and quantities of accessory equipment are included in the closed system CF, either integral to the isolators or external to them.

The following operational parameters have been considered and collected for the analysis, both from the open and closed system CFs: Annual energy supply for ventilation of the controlled environments (kWh);Annual energy supply for installed equipment (kWh);Twenty-one-day production cycle costs (in EUR);GHG emissions (in CO2eq);An estimation of building costs for both open and closed systems.

### 2.1. Annual Energy Supply for Ventilation of the Controlled Environments

The total air volume changes for each controlled environment in both open and closed system layouts have been calculated based on the different surface areas of the two CFs and the relevant Grade air changes per hour:-Grade B: 55 changes per hour;-Grade C: 35 changes per hour;-Grade D: 15 changes per hour.

### 2.2. Annual Energy Supply for Installed Equipment (kWh)

The power consumption was obtained from the Technical Department considering the kWh consumed from equipment installed both in Grade B and in Grade D, as detailed in Table 1.

### 2.3. Cost Analysis

A production cycle of 21 days/cycle was considered both in the open and closed systems.

These costs, provided by the Regina Margherita hospital, have been divided into indirect and direct costs as follows: 

#### 2.3.1. Indirect Costs for the Facility

-Lighting/equipment power;-Air facility/HVAC;-Environmental validation;-Annual qualification tools.

#### 2.3.2. Indirect Costs for the Production

-Media fill for two operators;-CF cleaning (biocides + clothing);-Qualified operator dressing qualification;-External operator dressing qualification;-Material flow;-Environmental controls (monthly).

#### 2.3.3. Direct Costs for the Production

-Biocides;-Dressing;-Staff (man/hour cost).

Reagent and material costs were not considered as specific to each production process.

### 2.4. GHG Emissions in (CO2eq)

The carbon footprint was obtained considering the characterization factor, a factor derived from a characterization model that is applied to convert an assigned life cycle inventory analysis result to the common unit of the impact category indicator (ISO 14040) for the production of 1 kWh in Italy (0.8), and multiplying by the electricity reduction for this characterization factor [10]. The calculated value corresponds to 406.31g/kWh. Such an approach is normally implemented in the life cycle assessment (LCA) study of products and is strongly recommended by the standard ISO 14067. This document specifies principles, requirements, and guidelines for the quantification and reporting of the carbon footprint of a product (CFP), in a manner consistent with International Standards on LCA, ISO 14040, and ISO 14044 [11].

### 2.5. Estimate of Infrastructure Building Costs 

The costs for infrastructure building are related to the complexity and size of the facility. The analysis has been made considering the range of the prices/m^2^ for this type of facility in Europe.

## 3. Results

### 3.1. Annual Energy Supply for Ventilation of the Controlled Environments

Based on the different surface areas of the two configurations and the relevant grade air changes, the total average daily energy consumption is 435 kWh/day at Regina Margherita CF and 63.23 kWh/day in the closed system, with 85% savings on ventilation costs, as shown in Table 2. With different external thermo-hygrometric conditions (i.e., summer weather vs. winter), given that the volumes of air treated are the same in both cases (open system and closed system), the difference between the emissions in the two cases will remain the same even if, in extreme conditions, the powers used could increase.

### 3.2. Annual Energy Supply for Installed Equipment (kWh)

The total installed power of the equipment is listed in Table 1 for each single configuration. The total installed power for open system equipment is 9.0 kW. In the case of the closed system, some instruments have been considered embedded in the three isolators used for this layout, and their installed powers are included in the total isolator power. The total installed power for the open system equipment is 11.6 kW, which is higher than the open system’s total installed equipment power, due to the higher number of isolators’ required power.

### 3.3. Cost Analysis

The production cycle of 21 days/cycle at the open system CF involves three internal operators and two external operators, as well as the quality control procedures.

Table 3 list the values in EUR for the 21-day production cycle provided by the open system at Regina Margherita CF.

For the closed system, an identical production cycle of 21 days has been considered, and the relevant direct and indirect costs have been calculated based on the values obtained from the equivalent open system CF in CPC (Epalinges), considering, whenever applicable, the differences (positives or negatives) compared to the open system values.

Table 4 below summarizes the costs for the closed system setting.

The closed system costs of the 21-day production cycle that have a lower value compared to those of the open system solution are the following: ⮚Air facility/HVAC;⮚Environmental validation;⮚Annual qualification tools;⮚Cell Factory cleaning (clothing and biocides);⮚Qualified operators dressing qualification;⮚External operators dressing qualification;⮚Biocides;⮚Dressing.

See Table 1 above.

As a matter of interest, it should be noted that the value of energy consumption for the lighting and equipment power is higher for the closed system CF compared to the open system: specifically, it is 13,671 kWh versus 7174 kWh/process, respectively. This is understandable, because the isolators used in the closed system are more complex equipment than the equivalent BSCs used in the open system CF and, therefore, require more energy. Nevertheless, despite this negative factor, the overall result for a 21-day production cycle is clearly in favor of the closed system solution.

### 3.4. GHG Emissions (in CO2eq)

The carbon footprint was obtained considering the characterization factor [10], applied to convert an assigned life cycle inventory analysis result to the common unit of the impact category indicator (ISO 14040) for the production of 1 kWh in Italy (406.31 g/kWh). The GHG emissions are illustrated in the Table 5 for the two configurations.

### 3.5. Estimate of Infrastructure Building Costs

The costs for infrastructure building are obviously related to the complexity and size of the facility. 

It is easy to understand that the higher volumes and higher classifications required for the environments of an open system CF will be a key factor in increasing the relevant building costs.

An open system layout will understandably have quite complex air distribution ducting and higher controlled environments volumes due to the need to have Grade D, C, and B environmentally controlled areas. All of these factors will increase the complexity of the infrastructure, and, therefore, the project and overall building costs of an open system CF compared to a closed system solution.

As a rule of thumb, without going into the details regarding the benchmark prices, in Europe, the range of the prices/m^2^ for this type of facility can be estimated as follows: 

Price/m^2^ open system facility: EUR 10,000 to EUR 15,000/m^2^;

Price/m^2^ closed system facility: EUR 5000 to EUR 8000/m^2^.

The data obtained, in terms of consumption and costs, are summarized in the following Figure 4.

## 4. Discussion

During the last few years, many studies have been published on the environmental footprint of different health care sectors. Methods of health technology assessment (HTA), such as economic evaluations and comparative effectiveness studies, allow one to determine whether the economic cost of a new pharmaceutical, medical device, or model of care is justified, given the health benefits the technology will afford to patients [12].

Recently, more than 200 leading health journals have published a joint commentary on the current climate emergency, with a call for urgent action to reduce the impact of climate change on health [13]. Carbon footprint modelling is a tool that has been applied to countries, institutions, industries, and individuals to determine the total Greenhouse Gas (GHG) emissions that are caused by an activity or product over its life stages [14]. Its application in the health sector is recent, but it has a large scope to aid policy makers, health care supply chain procurement, hospital managers, and clinical practitioners to develop novel solutions to reduce the climate change contributions of health care [15]. The carbon footprint is the unit of measurement of humanity’s demand for natural resources. This parameter, expressed as the carbon dioxide equivalent (CO2eq), normally used to estimate GHG emissions caused by products, services, organizations, events, and individuals [11], applies also to biological drugs production generated by complex processes that require compliance with guidelines and specialized personnel as well as important and impacting controlled contamination infrastructures.

While advanced therapy medicinal products (ATMPs) are revolutionizing the clinical treatment of many pathologies that are currently incurable, on the other hand, their preparation complexity, high production costs, and the necessary compliance with GMPs make them complicated and costly to use today. In this study, we have analytically calculated the carbon footprint of ATMP production by comparing the production cycles respectively produced in an open AinB system and in a closed AinD system, as defined by EudraLex Vol.4, Annex 1 and Part IV [8]. 

Because biological drugs cannot be sterilized, they require sterile environments for their manufacture. These environments are kept aseptic by a strict control of the environmental particulate and microbiological contamination obtained through ventilation systems, which, by filtering, recirculating, and conditioning the air in the environment, guarantee its sterility. The classic open system AinB clean room requires a high number of air recirculation, resulting in, therefore, a high energy consuming system. By contrast, the use of isolators can ensure a very high level of protection against the risk of product contamination and, at the same time, provide the operators with a very safe working environment. [6]. A Grade A isolator for the handling of biological drugs, installed in a Grade D environment, not only requires a much lower number of air changes but also, from an operational point of view, the savings obtained are substantial (about 85% energy reduction compared to an open system). 

The higher volumes and higher classifications required for the environment of an open system increase the initial building costs but also the maintenance costs. Although the isolators used in the closed system are more complex equipment, the overall result for a production cycle is clearly in favor of the closed system solution in terms of costs.

It is clear that one of the main problems of ATMP manufacturing is the high cost of production, which includes both direct and indirect costs. The direct costs are all those expenses regarding sanitary equipment, materials, and services directly involved in the production workflow (costs for the acquisition of primary materials, such as reagents, drugs, relative instruments, and appropriate clothing garments), personnel involved, and quality controls. For indirect costs, on the other hand, we refer to all the expenses sustained for materials, services, and maintenance necessary for the functioning of the production activity. Some examples include electrical implants, instrument validations, cleaning, and sanitization [16]. 

In general, new healthcare interventions are more expensive than the existing ones. Nevertheless, they usually provide added benefits over the standard of care. Thus, decisions makers (e.g., healthcare professionals, politicians, and other stakeholders) have to consider whether or not the new intervention is both affordable and an efficient use of limited resources [16,17].

We analyzed a hypothetic production cycle of 21 days/cycle, both in the open and closed systems, and highlighted a reduction in production costs in the isolator setting. 

Furthermore, by using an isolator in a Grade D room, the cost of disposable sterile clean room gowning is significantly reduced compared to the cost required by an open system; furthermore, decontamination cycles are designed and validated for much smaller volumes and automated (generally using hydrogen peroxide vaporization) and do not require specialized personnel. Even the validations of smaller environments used in the AinD closed system processes, compared to those performed in much larger AinB open systems—which, by definition, also include Grade B, C, and D environments—are substantially less expensive and time consuming.

Although the use of closed systems is long established in pharmaceutical settings, the manufacturing of advanced therapies (ATMPs) is still anchored to the logic of the world of research and adopts processes carried out in clean rooms or open systems [18]. Moreover, the growth, selection, and modification of ATMPs may require the use of multiple equipment, which is not always manageable within closed systems. However, a properly designed closed system allows a high level of flexibility for managing different processes in the same environment or even simply allows the movement of the ATMP manufacturing plants to different sites with respect to the static nature of the open system [6].

In this study, we have analyzed the sustainability of a classic clean room comparable to a closed system equipped with isolators, in terms of Greenhouse Gas (GHG) emissions expressed in CO2eq, to highlight opportunities for improving the environmental impact of ATMPs manufacturing. 

The actual energy consumption data of two ATMP production processes with equal production outputs were collected. We compared an open AinB system (using laminar flow Grade A cabinets and other necessary equipment installed in Grade B clean rooms) and an AinD closed system production unit (using Grade A isolators with embedded necessary equipment installed in Grade D clean rooms). The energy consumption calculations also take into account the consumption of the individual equipment used in the open system, such as the laminar flow cabinet, incubator, centrifuge, etc. 

This analysis shows that the energy savings are 6497 kWh/21 days of the production process and that, consequently, the reduction of GHG emissions using closed systems is in the range of 52% per process cycle.

Paradoxically, but understandably, the energy consumption of the Grade A isolator used in the closed system CF is higher than the energy used by the BSCs installed in the Grade B open system for the Grade A critical area (13,671 kWh versus 7174 kWh/process, respectively). 

However, comparing the total value of energy consumption calculated as the sum of the energy required for ventilation, distribution and air treatment, and instrumentation, we have demonstrated that the energy required to complete a 21-day production cycle in an open system is considerably higher than that required by an equivalent production cycle in a closed system. The same applies to GHG emissions, which are higher for the open systems compared to those generated by the use of closed systems. 

A look to the future for ATMP could be represented by prefabricated, modular, and mobile clean rooms. The construction of a clean room generally requires very high construction times and costs, and it requires availability in the existing structures (or in those of the project) of considerable volumes to be dedicated to the construction of the plant. Modular units have been developed to address infrastructure needs for any emergency, such as the recent COVID-19 pandemic [19].

A prefabricated structure equipped with one or more insulators for the production of ATMPs represents a great simplification compared to the classic realization of controlled-atmosphere environments, allowing institutions and/or companies to have a production area for advanced therapies very usable in a single step and in a short time. This allows for a low-cost Point of Care designed as a solution that can be easily installed in outdoor spaces, car parks, lawns, or hospitals where the production of ATMPs will be implemented. The modular prefabricated clean room contains building blocks specifically designed to create a permanent Point of Care and low-cost Grade D clean rooms fitted with isolators and capable of being installed in the outdoor spaces of hospitals or treatment research centers. The facility is therefore equipped with all the necessary equipment for ATMP production in a regulated environment that complies with GMP recommendations. The building block design allows for complete customization of the structure, from the shape of the changing room to a multi-space layout that can include more than one clean room. Furthermore, the structure can easily be expanded or modified after installation by adding new customized construction elements. The installation of the prefabricated structures is very simple and does not require pre-installed structures or a connection to hospital buildings (see Figure 1B).

## 5. Conclusions

ATMPs require major investments in all stages of development, from preclinical and clinical trials to manufacturing and commercialization.

The hospital can often encounter significant obstacles during the clinical development of ATMPs. This is largely related to the lack of financial support and logistical or engineering difficulties. However, public institutions play an important role in innovation and scientific expertise for the development of ATMPs. In a hospital setting, to concretely guarantee patients’ access to these innovative therapies, it is of fundamental importance that advanced therapies are considered an investment for the Health Service, as ATMPs represent an extraordinary example of how biopharmaceutical research is capable of expressing the great potential of the convergence between science, technology, and clinical practice.

The challenge is to reach a new health approach where researchers and private companies could share experience and resources for the application of new targeted and personalized therapies, such as ATMPs.

## Figures and Tables

**Figure 1 bioengineering-10-01047-f001:**
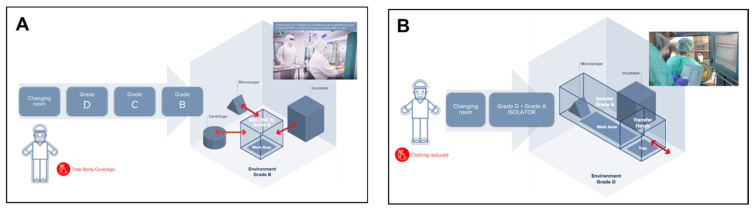
ATMP production systems as defined by EU GMP Guidelines. (**A**) Open system, (**B**) closed system.

**Figure 2 bioengineering-10-01047-f002:**
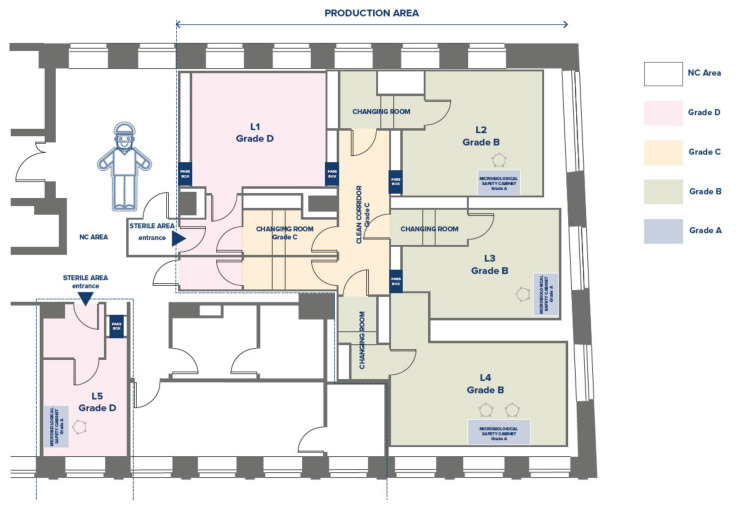
Cell Factory layout at Regina Margherita Children’s Hospital, City of Health and Science of Turin.

**Figure 3 bioengineering-10-01047-f003:**
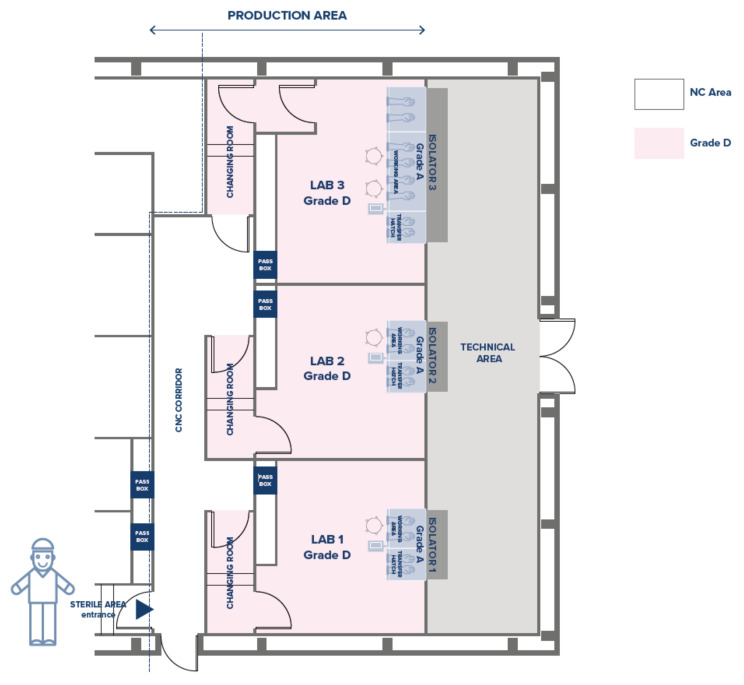
Closed system layout with equivalent production output.

**Figure 4 bioengineering-10-01047-f004:**
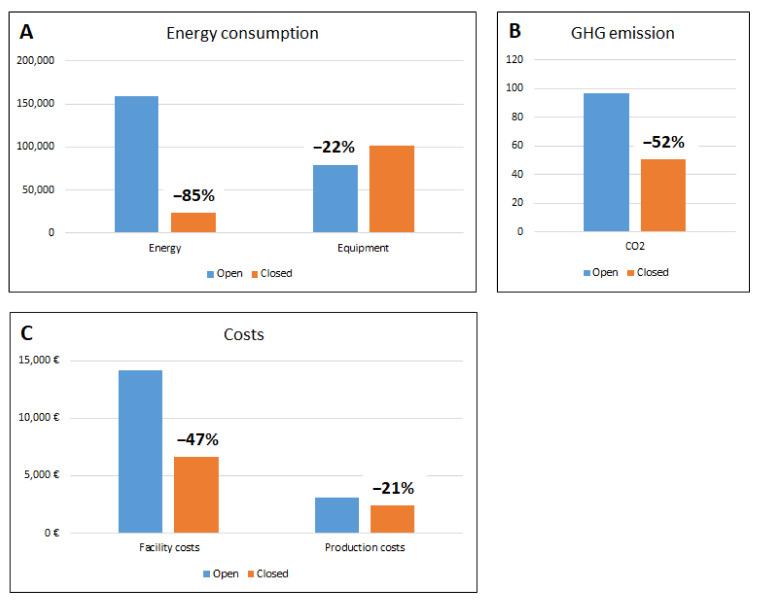
Summary of results.

**Table 1 bioengineering-10-01047-t001:** Installed equipment powers.

LAB	Equipment	Quantity	Equipment Power (kW)	Total Installed Power (kW)
**Open System**				
L2	1.20 mt BSC	1	0.3	0.3
L2	CO_2_ Incubator	1	1.5	1.5
L2	Centrifuge	1	1	1
L3	1.20 mt BSC	1	0.3	0.3
L3	CO_2_ Incubator	1	1.5	1.5
L3	Centrifuge	1	1	1
L1	Fridge	1	0.3	0.3
L4	1.80 mt BSC	1	0.6	0.6
L4	CO_2_ Incubator	1	1.5	1.5
L4	Centrifuge	1	1	1
**Total Installed Equipment Power (kW)**				**9**
**Total Installed Equipment Annual Consumption (kWh)**				**78,840**
**Closed System**				
LAB 1	Two-glove isolator	1	2.2	2.2
LAB 1	CO_2_ Incubator	embedded		0
LAB 1	Centrifuge	1	1	1
LAB 2	Two-glove isolator	1	2.2	2.2
LAB 2	CO_2_ Incubator	embedded		0
LAB 2	Centrifuge	1	1	1
LAB 2	Fridge	1	0.3	0.3
LAB 3	Four-glove isolator	1	4.9	4.9
LAB 3	CO_2_ Incubator	embedded		0
LAB 3	Centrifuge	embedded		0
**Total Installed Equipment Power (kW)**				**11.6**
**Total Installed Equipment Annual Consumption (kWh)**				**(kWh) 101,616**

**Table 2 bioengineering-10-01047-t002:** Annual HVAC energy for the controlled environments.

	Open System	Closed System
Controlled Environments	3 Grade B Production Areas + D and C Areas	3 Grade D Equivalent Production Areas
Total controlled environments surface (m^2^)	107	45
Total controlled environments volume (m^3^)	271	108
Total recirculated air volume/hour (m^3^/h)	11,247	1620
Average air changes/h	41	15
Daily energy kWh	435	63.23
Total annual energy kWh	158,775	23,079

**Table 3 bioengineering-10-01047-t003:** Indirect and direct costs for a 21-day cycle run in the open system CF.

Indirect Costs	Costs (EUR)
**Facility:**	
Lighting/power equipment	EUR 436.38
Air facility/HVAC	EUR 2503.87
Environmental validation	EUR 1516.22
Annual qualification tools	EUR 4811.24
**Production:**	
Media fill for two operators	EUR 2058.12
Cell Factory cleaning (clothing + biocides)	EUR 1715.77
Qualified operator dressing qualification	EUR 247.02
External operator dressing qualification	EUR 546.38
Material flow	EUR 230.78
Environmental controls (monthly)	EUR 112.02
**Total indirect costs**	EUR **14,177.80**
**Direct Costs**	
**Production:**	
Biocidal products	EUR 333.35
Dressing	EUR 176.40
Total material cost	EUR 509.75
Staff (34h30)	EUR 2058.00
**Total direct costs**	EUR **3077.50**
**Total indirect and direct costs**	EUR **17,255.30**

**Table 4 bioengineering-10-01047-t004:** Indirect and direct costs for a 21-day cycle run in the equivalent closed system.

Indirect Costs	Costs (EUR)
**Facility:**	
Lighting/power equipment	EUR 631.89
Air facility/HVAC	EUR 350.54
Environmental validation	EUR 631.76
Annual qualification tools	EUR 2004.68
**Production:**	
Media fill for two operators	EUR 2058.12
Cell Factory cleaning (clothing + biocides)	EUR 490.22
Qualified operator dressing qualification	EUR 98.81
External operator dressing qualification	EUR 218.55
Material flow	EUR 75.15
Environmental controls (monthly)	EUR 40.00
**Total indirect costs**	EUR **6599.72**
**Direct Costs**	
**Production:**	
Biocidal products	EUR 111.12
Dressing	EUR 70.56
Total material cost	EUR 181.68
Staff (34h30)	EUR 2058.00
**Total direct costs**	EUR 2421.36
**Total indirect and direct costs**	EUR **9021.08**

**Table 5 bioengineering-10-01047-t005:** Yearly GHG emissions in CO2eq (metric tons).

	HVAC Energy Consumption for Controlled Environments (kWh)	Equipment Energy Consumption (kWh)	Total Energy Consumption (kWh)	CO2eq Metric Tons
	Annual	Annual	Annual	Annual
Open system	158,775	78,840	237,615	96.55
Closed system	23,079	101,616	124,695	50.66

## Data Availability

The datasets used and analyzed for the current study are available from the corresponding author upon reasonable request.

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
