# Peer review of "Evaluating Ecological Impact and Sustainability in the Manufacturing of Advanced Therapies: Comparative Analysis of Greenhouse Gas Emissions in the Production of ATMPs in Open and Closed Systems"

_bioengineering, 2023, doi:10.3390/bioengineering10091047_

Round 1

Reviewer 1 Report

The article aims to analyze and compare greenhouse gas emissions from ATMP production performed in open and closed systems with equivalent performance. This is a rather urgent task, the solution of which can give a direction for reducing costs in the production of ATMP. The authors have done a good job, but there are comments that should be corrected before publication. Therefore, I recommend the major revision.

Key comments:

1. The part of the Introduction regarding the description of the difference between open and closed systems, including Figures 1 and 3, should be moved to the Materials and Methods section, and the part regarding the relevance of this study should be improved in the Introduction section.

2. It is not clear from the article whether emissions for open and closed systems will differ during the year. Is there an influence of the temperature and humidity of the outside air?

3. Part of the discussion should be improved while increasing the number of references.

4. The article does not disclose the complexity of using closed systems.

5. There are typos in table 1: Kw instead of kW, there is no consumption dimension in case of a closed system. There is no equipment run time, so it is not possible to estimate the total electricity consumption. Decimal separator - comma? Consumption 78.84 or 78,840 kWh? For what period of time? TOTAL INSTALLED EQUIPMENT POWER is measured in (kW).

7. In tables 3 and 4 it is necessary to explain "Staff (34,3h)". Is this the duration of the staff? Decimal separator - comma?

7. In table 2, the "m2" is applied, and on lines 328-329 - "sqm".

8. In the Results section, you should graphically present the obtained data.

Author Response

We wish to thank you for giving us the opportunity to revise and improve our work.

In this revised version, we took into consideration the reviewers’ criticisms and modified the text accordingly, as explained below.  All the revisions are tracked in red in the text.

All authors actively participated in the revision of the manuscript, and approved its submission.

We hope that the manuscript will be now considered worthy of publication.

Kind regards,

Ivana Ferrero

Reviewer 2 Report

I have had the pleasure of reviewing your manuscript titled " Ecological Footprint and Sustainability of Advanced Therapies manufacturing: a Systematic Analysis and Comparison between Greenhouse Gas Emissions of ATMPs production, performed in Open Systems and Closed Systems with an equivalent production output." I must commend your comprehensive and robust approach to this pertinent issue in the realm of Advanced Therapy Medicinal Products (ATMP) manufacturing.

However, there are some crucial amendments required as follow:

Point 1: The suggested title of the MS should represent accurately the performed experiments without unnecessary information. I would suggest the title to be “Evaluating Ecological Impact and sustainability in the Manufacturing of Advanced Therapies: Comparative Analysis of Greenhouse Gas Emissions in the Production of ATMPs in Open and Closed Systems."

Point 2: My second comment pertains to the abstract section, where authors should provide a concise overview of the research topic, objectives, and methods employed. The current abstract lacks sufficient details about the actual findings and outcomes of the study. The importance of transparently sharing detailed results in the abstract is crucial to facilitate the broader dissemination, interpretation, and meaningful engagement with the research.

Point 3: The meaning is not clear for the reader from “They have a very high potential” in line 33-34. Please replace with “These products are utilized to restore, correct, or modify physiological functions primarily through a pharmacological action" [1]. ATMPs hold significant potential for a wide range of medical conditions and have the capacity to benefit millions of patients. These products are categorized into four primary groups:

Point 4: My second comment pertains to line 189-191 where you illustrated the main objective of the study. The hypothesis of your work is missed justify with detailed hypothesis.

Point 5: The  carbon dioxide chemical symbol CO2 needs to be double checked through the MS as well as in Tables 1 and 5.

Point 6: My next comment pertains to conclusion section, by which should the conclusions align with the research objectives and questions outlined in the introduction. The conclusion should directly address the main research goals and hypotheses. Please rewrite this section providing more details according to the main findings.

Point 7: While your manuscript displays a strong command of the topic and presents compelling findings, I noticed some major language issues and inconsistencies throughout the text. These could potentially hinder the clarity of your message and disrupt the reader's engagement with your work.

I noticed some major language issues and inconsistencies throughout the text. These could potentially hinder the clarity of your message and disrupt the reader's engagement with your work.

Author Response

(The authors gave the same response as above.)

Round 2

Reviewer 1 Report

The authors have done a good job of eliminating major comments, so I recommend accepting this article as is.

Reviewer 2 Report

Suggested amendments have been performed. Best Wishes